# Cardiometabolic Traits in Adult Twins: Heritability and BMI Impact with Age

**DOI:** 10.3390/nu15010164

**Published:** 2022-12-29

**Authors:** Xuanming Hong, Zhiyu Wu, Weihua Cao, Jun Lv, Canqing Yu, Tao Huang, Dianjianyi Sun, Chunxiao Liao, Yuanjie Pang, Zengchang Pang, Liming Cong, Hua Wang, Xianping Wu, Yu Liu, Wenjing Gao, Liming Li

**Affiliations:** 1Department of Epidemiology and Biostatistics, School of Public Health, Peking University, Beijing 100191, China; 2Qingdao Center for Disease Control and Prevention, Qingdao 266033, China; 3Zhejiang Center for Disease Control and Prevention, Hangzhou 310051, China; 4Jiangsu Center for Disease Control and Prevention, Nanjing 210008, China; 5Sichuan Center for Disease Control and Prevention, Chengdu 610041, China; 6Heilongjiang Center for Disease Control and Prevention, Harbin 150090, China; 7Peking University Center for Public Health and Epidemic Preparedness & Response, Peking University, Beijing 100191, China

**Keywords:** twin study, body mass index, lipids, glycemic index, blood pressure

## Abstract

Background: The prevalence of obesity and cardiometabolic diseases continues to rise globally and obesity is a significant risk factor for cardiometabolic diseases. However, to our knowledge, evidence of the relative roles of genes and the environment underlying obesity and cardiometabolic disease traits and the correlations between them are still lacking, as is how they change with age. Method: Data were obtained from the Chinese National Twin Registry (CNTR). A total of 1421 twin pairs were included. Univariate structural equation models (SEMs) were performed to evaluate the heritability of BMI and cardiometabolic traits, which included blood hemoglobin A1c (HbA1c), fasting blood glucose (FBG), systolic blood pressure (SBP), diastolic blood pressure (DBP), total cholesterol (TC), triglycerides (TGs), low-density lipoprotein cholesterol (LDL-C) and high-density lipoprotein cholesterol (HDL-C). Bivariate SEMs were used to assess the genetic/environmental correlations between them. The study population was divided into three groups for analysis: ≤50, 51–60, and >60 years old to assess the changes in heritability and genetic/environmental correlations with ageing. Results: Univariate SEMs showed a high heritability of BMI (72%) and cardiometabolic traits, which ranged from 30% (HbA1c) to 69% (HDL-C). With age increasing, the heritability of all phenotypes has different degrees of declining trends. Among these, BMI, SBP, and DBP presented significant monotonous declining trends. The bivariate SEMs indicated that BMI correlated with all cardiometabolic traits. The genetic correlations were estimated to range from 0.14 (BMI and LDL-C) to 0.39 (BMI and DBP), while the environmental correlations ranged from 0.13 (BMI and TC/LDL-C) to 0.31 (BMI and TG). The genetic contributions underlying the correlations between BMI and SBP and DBP, TC, TG, and HDL-C showed a progressive decrease as age groups increased. In contrast, environmental correlations displayed a significant increasing trend for HbA1c, SBP, and DBP. Conclusions: The findings suggest that genetic and environmental factors have essential effects on BMI and all cardiometabolic traits. However, as age groups increased, genetic influences presented varying degrees of decrement for BMI and most cardiometabolic traits, suggesting the increasing importance of environments. Genetic factors played a consistently larger role than environmental factors in the phenotypic correlations between BMI and cardiometabolic traits. Nevertheless, the relative magnitudes of genetic and environmental factors may change over time.

## 1. Introduction

Cardiometabolic diseases are a severe public health burden and the leading cause of death, the prevalence of which continues to rise across global populations [1]. Obesity, as a major independent risk factor for cardiovascular morbidity and mortality, influences billions of people worldwide and is predicted to affect 20% of the global population by 2025 [2,3]. Both genetic and environmental factors contribute to the aetiology of cardiometabolic diseases and obesity [4]. Although these phenotypes have been proven to be highly heritable, genetic factors could not fully explain the growing prevalence worldwide of the diseases, suggesting that environmental contributions are of increasing importance [5,6,7]. In addition to genes and environments, the incidences of these phenotypes are also driven by the effects of age [8]. A clear understanding of the relative magnitudes of the genetic and environmental contributions underlying these phenotypes and how they change with age has considerable implications for dissecting major effectors in individuals and designing obesity and cardiometabolic disease control strategies.

Ageing and obesity are the major risk factors for many cardiometabolic diseases. Although previous studies have shown genetic and environmental correlations between several cardiometabolic traits and body mass index (BMI) [9,10,11], few investigations have explored the changes in genetic and environmental contributions underlying both BMI and cardiometabolic traits with ageing. Twin study design provides a valuable tool for evaluating these contributions by comparing the similarity between monozygotic and dizygotic twins over different age groups.

The objective of the current study was based on the Chinese National Twin Register (CNTR) to investigate the following questions: (1) the relative effects of genetic and environmental factors on BMI and cardiometabolic traits and the changes in different age groups; and (2) the degree to which genetic and environmental influences are shared on cardiometabolic traits and BMI and the changes in the degrees with ageing. 

## 2. Method

### 2.1. Study Population

The study population was extracted from the CNTR. The design, data collection procedures, and population characteristics of the CNTR have been detailed elsewhere [12]. After large-scale baseline and follow-up investigations, 61,566 twin pairs were recruited from 11 provinces/cities in China.

The data of this study were collected from questionnaire investigations (including demographics, medication history, and lifestyle), physical examinations (blood pressure, BMI, height, and weight), and blood tests (biochemical tests). Ethical approval of the present study was granted by the Biomedical Ethics Committee at Peking University, Beijing, China. Written informed consent from all study participants was provided. Blood samples were obtained from 3038 twins in CNTR who completed at least one follow-up survey between 2013 and 2018. Eligibility criteria included (1) twin members both available for physical examinations and had provided complete information on demographic characteristics and medical history, and (2) blood samples had been donated from both twin members. Exclusion criteria included rear apart/opposite-sex twin pairs. Twins who lived apart for more than a year before the age of five years were defined as rear apart. In total, 2842 participants (1421 twin pairs) were included.

### 2.2. Definition and Measurement of BMI, Cardiometabolic Traits, and Zygosity

Trained health professionals measured weight and height. BMI was calculated using weight divided by height squared (kg/m^2^).

Cardiometabolic traits in this study consisted of glycemic traits, including glycated hemoglobin (HbA1c) and fasting blood glucose (FBG); blood pressure traits, including systolic blood pressure (SBP) and diastolic blood pressure (DBP); and lipid-related traits, including total cholesterol (TC), plasma triglyceride (TG), low-density lipoprotein cholesterol (LDL-C) and high-density lipoprotein cholesterol (HDL-C). Glycemic traits and lipid parameters were measured after the participants fasted for at least 12 h. The HbA1c level was evaluated using high-performance liquid chromatography analysis, and the FBG level was determined by a modified hexokinase enzymatic method. The enzymatic colorimetric method was applied to detect TC and TG levels, and the direct approach was used for LDL-C and HDL-C. Outliers (>3 SD from the mean) were removed in the statistical analysis. SBP and DBP were measured twice in the sitting position after taking a rest of at least 5 min using an OMRON HEM-7200 electronic sphygmomanometer on the right arm. The two sequential measurements were averaged. A third measurement was obtained if the difference between the two measurements was >10 mmHg, and the two closest measurements were averaged. For participants who were using antihypertensive medication, the SBP and DBP levels were adjusted by adding 15 and 10 mmHg, respectively [13].

Twin zygosity was primarily determined based on a questionnaire method. For twins who shared the same gender, a model was established according to age, gender, and the answers to the question “Are they alike in appearance and difficult to distinguish?”, and were classified as monozygotic twins if they responded ‘Yes’. The accuracy of this method has been validated in a subgroup of CNTR, which revealed a high accuracy of 85 to 89% agreement with genetic markers [14,15].

### 2.3. Statistical Analysis

Statistical analysis was conducted according to age groups (≤50, 51–60, and >60 years). All twin analyses were performed using the R package “OpenMx” [16]. Statistical significance was set at *p* < 0.05.

### 2.4. Heritability Analysis

We used structural equation models (SEMs) to estimate the heritability of BMI and cardiometabolic traits. Detailed methods for the univariate structural equation model in the twin study have previously been described [17]. In short, the model allows the decomposition of the phenotypic variation (i.e., variance) into four parts: additive genetic variance (the cumulative effect of alleles) (A), nonadditive genetic variance (an effect caused by the interaction between alleles) (D), common environmental variance (variance due to environmental factors shared within twin pairs) (C), and unique environmental variance (variance due to environmental factors not shared within twin pairs) (E). In the univariate SEM, variance components and means were allowed to vary across age groups. Since the C and D effects are not easily discriminated in twins raised together [18], C or D is generally selected for model fitting together with A and E. The ACE model was used if the correlation between MZ twins was less than twice that of DZ. Otherwise, the ADE model was used.

Taking the ACE model as an example (Figure 1), as the MZ twins shared 100% of their genome sequences, the difference between them is attributed to unique environmental variance (E). Therefore, the correlation between MZ twins provides a raw estimation of (A + C). In contrast, DZ twins share an average of 50% of their genes. The correlation between DZs is thus a direct estimate of (½A + C). Significance tests of variance components (A and C) were performed by comparing saturated models with submodels that constrain the effects of pathways from latent variables to traits (path a, c) to 0 (Figure 1). As variance E includes measurement error and thus cannot be constrained to 0, the significance test for E is not conducted. The total variance was constrained to 1 for these models to ensure proper identification of each component. Based on these estimations, heritability is an approximation of the relative contributions of additive genetic variance [19]. We then conducted homogeneity models to test whether variance components can be equal across age groups. If the results of homogeneity models showed a poor fit, variance components were suggested to have statistically significant changes with ageing.

The best model fit was selected based on *p* values and Akaike’s information criterion (AIC).

### 2.5. Genetic and Environmental Correlation Analysis

For each age group, we conducted bivariate SEM to determine the degree to which shared genetic or environmental factors drove the correlations between BMI and cardiometabolic traits. Details of this model have been described previously (Figure 2) [17]. Similar to univariate SEM, the A, C, D, and E variations in BMI and cardiometabolic traits were decomposed. Then, the genetic and environmental contributions to the observed covariance between BMI and cardiometabolic traits were identified by using a series of submodels, which assessed whether genetic, shared environmental, and unique environmental pathways from BMI to cardiometabolic traits can be constrained to 0. We only performed the ACE models in this study because univariate analyses did not support that ADE models were better fitted in studying BMI and cardiometabolic traits.

Logarithmic transformation was performed for variables with a non-normal distribution. Since gender and age were influencing factors for both BMI and cardiometabolic traits, and subtle differences in age in each age group may cause confounding, the structural equation models were all adjusted for age and sex. The models that involve glycemic traits and lipid-related traits were further adjusted for the use of glucoregulatory and lipid medication.

## 3. Results

### 3.1. Characteristics of the Study Population

The demographics and characteristics of the participants are detailed in Table 1. Among the 2842 participants, 1690 were MZ twins, and 978 were females. Among these, 1684 participants were from 18 to 50 years, 702 were from 51 to 60 years, and 456 were over 60 years, with 942, 438, and 306 MZ twins, respectively.

### 3.2. Heritability for BMI and Cardiometabolic Traits

We first performed univariate SEM analysis in the full analysis population. The fitting procedures are shown in Appendix A. In short, for BMI and cardiometabolic traits, ACE models were generally the better fitting models than ADE, and in these models, the shared environmental parts (C) were not significant for every phenotype (set as 0). The estimates of heritability were 0.72 for BMI, 0.30 for HbA1c, 0.49 for FBG, 0.39 for SBP, 0.39 for DBP, 0.63 for TC, 0.58 for TG, 0.60 for LDL-C, and 0.69 for HDL-C in the full population (Table 2). To explore the potential differences between sex in heritability, we further conducted the subgroup analyses according to sex, and no noticeable sex differences were observed in our study (P_homogeneity_ values > 0.05, Appendix A).

Details on the univariate analysis procedures are provided in Appendix A by age group. The point estimates of MZ twin correlations in all age groups were higher than those of DZ twins for all phenotypes, indicating that BMI and cardiometabolic traits were consistently affected by genetic factors (Figure 3).

The estimations of heritability for BMI and cardiometabolic traits by age group are presented in Figure 4. Compared with the first age group, heritability showed different degrees of decline in the latter two groups for all phenotypes. The variance components could not be equated across age groups (*p* values < 0.01), suggesting an alteration of heritability with ageing for these traits. (Appendix A). The pattern of monotonic decline with age was presented for the heritability of BMI, FBG, SBP, DBP, and TG. Significant decreasing trends in heritability were observed for BMI, HbA1c, and SBP: from 0.74 to 0.60 for BMI, from 0.67 to 0.33 and 0.34 for HbA1c, 0.61 to 0.42 for SBP, with *p* values < 0.01 in homogeneity models. The remaining segment of the variation for BMI and cardiometabolic factors was ascribed to environmental influences.

### 3.3. Genetic and Environmental Factors Underlying the Correlations between BMI and Cardiometabolic Traits

We performed bivariate SEM analysis for all cardiometabolic traits to estimate the relative contributions of genetic and environmental factors to their correlations with BMI in the full analysis population. The fitting procedures are presented in Appendix A. AE models were generally the best fit for BMI and cardiometabolic traits in the univariate SEM, which was also corroborated in this part of the analysis. BMI was phenotypically correlated with all cardiometabolic traits. The results of the bivariate SEM are shown in Table 3. BMI shared the genetic components with all cardiometabolic traits, and the estimates of genetic correlations were 0.23 and 0.18 for HbA1c and FBG, 0.31 and 0.39 for SBP and DBP, and 0.19, 0.37, 0.14, and −0.34 for TC, TG, LDL-C, and HDL-C. Environmental contributions also accounted for all the associations between BMI and cardiometabolic traits, with point estimates of 0.14 and 0.17 for HbA1c and FBG, 0.17 and 0.20 for SBP and DBP, and 0.13, 0.31, 0.13, and −0.15 for lipid measures. In summary, the decomposition of the phenotypic correlations implied that A and E variations are both significant for the correlations of BMI and cardiometabolic traits, suggesting the correlations can be explained by shared genetic and unique environmental factors. Nevertheless, they were primarily (59 to 80%) due to genetic correlations. In addition, a subgroup analysis by gender was conducted, and for all the correlations between BMI and cardiometabolic traits, homogeneity tests indicated significant differences between gender (Appendix A).

The statistical procedures and results of bivariate SEM by age group are shown in Appendix A, and Figure 5. For all cardiometabolic traits, the point estimations of their genetic correlations with BMI showed different degrees of reduction compared to the age group of ≤50. Genetic correlations of BMI with SBP, DBP, TC, TG, and HDL-C showed progressive decreases in point estimates with ageing (Figure 5). Among these, SBP and DBP demonstrated significant declining trends: from 0.44 to 0.08 for SBP and from 0.49 to 0.00 for DBP with nonoverlapping confidence intervals, and the variance could not be equated across age groups (Appendix A). In contrast, the point estimations of environmental correlations displayed an increasing trend for HbA1c, SBP, and DBP, indicating that the role of environmental contributions is becoming more critical.

## 4. Discussion

In the current study, we report a high heritability of BMI (72%) and cardiometabolic traits, which ranged from 30% (HbA1c) to 69% (HDL-C) in the full analysis population. We then assessed the changes in heritability with increasing age. BMI and all cardiometabolic traits showed declines in heritability but to varying degrees. Among these phenotypes, BMI, SBP, and DBP presented significant declining trends. BMI was correlated with all cardiometabolic traits in this study. By using bivariate SEM analysis, the genetic correlations were estimated to range from 0.14 (BMI and LDL-C) to 0.39 (BMI and DBP), and the environmental correlations behind the associations ranged from 0.13 (BMI and TC/LDL-C) to 0.31 (BMI and TG). As the age increased, the genetic contributions underlying the correlations between BMI and SBP, DBP, TC, TG, and HDL-C showed a progressive decrease in point estimates. In contrast, environmental correlations displayed a significant increasing trend for HbA1c, SBP, and DBP.

### 4.1. Heritability of BMI and Cardiometabolic Traits

Our estimate of the heritability of BMI is similar to the heritability (70–80%) reported from twin studies in a systematic review by Patel et al. [6]. In accordance with the current analysis, another study that included 40 twin cohorts from 20 countries observed that the heritability of BMI decreased over age: the heritability of BMI decreased from 0.77 and 0.75 in 20- to 29-year-olds to 0.57 and 0.59 in 70- to 79-year-olds for males and females, respectively [20]. Genetic and environmental factors both play essential roles in the development of obesity [21,22]. One possible explanation for the decreasing heritability of BMI was that the unique environmental factors present increasing effects with ageing. A recent family-based study on BMI-related gene-environment interactions showed that environmental factors involving age, alcohol consumption, and dietary habits might interact with BMI-related genes, suggesting that environmental effects could alter with age [23]. However, there are studies on BMI heritability change over time that yielded inconsistent conclusions. A review of sixteen studies reported that the heritability of BMI remains consistently high between age categories but demonstrates no significant difference in changes [24]. A study enlisted 235 monozygotic twin pairs and 260 dizygotic twin pairs from the United States military. After three clinical exams from the participants, an increase over time was observed in the heritability of BMI (from 0.48 to 0.61) in this study [25]. The heterogeneity of previous heritability estimates may be explained by different age ranges between studies [26,27]. A genome-wide association study on adult body size identified 15 loci with significant age-specific effects. Among these, 11 considerably impacted adults less than 50 years old [28]. Another explanation may be the difference in race and study design because the expression of BMI-related genes differed between populations [26,29].

On the other hand, our heritability estimates for the cardiometabolic traits are in agreement with the majority of earlier reports [5,9,30,31,32]. Few studies have investigated the changes in heritability over time for glycemic and lipid-related traits. A study based on the Danish Twin Registry found that the heritability for 120 min blood glucose decreased with age in women and fasting insulin in men, indicating age differences for the variance components [33]. However, this study also found that the heritability for 120 min blood glucose was increased in males, suggesting that the interpretation of the effects of age on genetic and environmental factors for glycemic traits is not straightforward. Our findings indicated that the heritability of blood glucose may remain essentially unchanged.

For blood lipid-related traits, a longitudinal study that included 415 adult twins reported that the heritability of HDL, LDL, and TG might vary with ageing, indicating that genetic factors might influence components of cholesterol and triglycerides differently over time. Nevertheless, no significant trends were observed [34]. The other relevant studies mainly focused on adolescents, consistently concluding that the genetic effects on lipid-related traits change over time [35,36]. A previous study found that the expression of fatty acid binding protein 3 (*FABP3*) is age-related, and it modulates lipid remodeling and subsequent muscle homeostasis, which is in support of our conclusions that with aging, the influence of genes on lipid traits may vary [37].

Regarding SBP and DBP, several studies have reported that the heritability of SBP and DBP estimates tends to decrease with age [38,39]. Gene–age interactions for blood pressure have been reported, suggesting that the effect of genetic factors on blood pressure varies by age [40]. A genome-wide association study that involved 99,241 individuals of European ancestry identified 20 loci with age-dependent effects on blood pressure-related traits. Nine of these loci demonstrated age-dependent effects on blood pressure by examining the interactions alone [41].

### 4.2. Correlations of BMI and Cardiometabolic Traits

Since BMI and cardiometabolic traits are both highly heritable traits, an interesting question is to what degree the correlations between BMI and cardiometabolic traits are driven by the underlying genetic/environmental factors in common to these traits and how the degrees changed over time. To answer this question, we conducted a bivariate twin analysis on BMI and cardiometabolic traits. The genetic and environmental correlations were revealed to be modest or moderate correlated but were all significantly different from 0. The genetic correlations ranged from 0.18 to 0.23 for glycemic-related traits, 0.31 to 0.39 for blood pressure, and 0.21 to 0.35 for lipid-related traits, which were similar to previous reports from different ethnic groups [9,42,43]. These results indicate a limited but not negligible overlap between genetic factors associated with both BMI and cardiometabolic traits, suggesting the role of pleiotropic genetic mechanisms underlying the correlations between BMI and cardiometabolic traits across populations. A study based on two large GWAS data sets, including 339,224 subjects from a BMI data set and 110,452 subjects from a glycemic data set, reported that 5 182 SNP loci showed pleiotropy for both phenotypes [44]. For blood pressure, several relevant studies have concluded that obesity-related genes can explain part of the genetic variation in blood pressure [45]. A study performed a bivariate genome-wide association study for BMI and blood pressure in the northern Chinese twin population and identified 27 quantitative trait nucleotides (QTNs) for BMI and blood pressure [46]. For the lipid phenotypes, evidence of the development of obesity and dyslipidemia shared many genetic pathways involved in adipogenesis, lipid metabolism, and energy homeostasis that have been previously reported [47]. 

Weak to modest but statistically significant environmental correlations were observed for these associations in our study. Similar results were concluded in several previous studies [9,48]. These findings suggested that the relationship between BMI and cardiometabolic traits was partly explained by shared genes and partly by environmental effects. However, there remains contention on the role of environmental contributions between BMI and these phenotypes [10,11]. Although the specific mechanism by which BMI affects the genetic and environmental contributions of cardiometabolic traits cannot be determined in this study, it is well known that several environmental and lifestyle factors, such as unhealthy diets and lack of physical activity, influence the risk of both obesity and cardiometabolic disease [49]. In addition, we also reported that there were significant sex differences for the genetic and environmental contributions underlying the correlations.

A novel finding in this study is that genetic correlations with BMI may progressively decrease with ageing for SBP, DBP, TC, TG, and HDL-C. In contrast, the environmental correlations displayed an increasing trend for HbA1c, FBG, SBP, DBP, and HDL-C over time. This situation may have arisen due to the enhancement of the environmental effect relative to the genetic effect. Relevant research is limited. Consistent with our findings, a longitudinal study focused on 341 male and 292 female twins aged 20–50 years and performed tests at two time points suggested that the unshared environmental factors for twin pairs may contribute more strongly to the increased risk of glycemic deterioration [50]. Similar, another study involved 284 South Korean twin individuals and 279 non-twin family members with 3.1 year changes in adiposity traits and metabolic syndrome traits data. Its results from bivariate SEM analysis indicated that environmental contributions play a more critical role in the correlations between changes in BMI and changes in DBP, TG, and HDL-C [51].

To our knowledge, this is the first study to reveal the changes in genetic and environmental factors underlying the correlations between BMI and multiple cardiometabolic traits with age. Our results illustrated that the role of environmental factors becomes increasingly essential and emphasized the significance of environmental intervention in preventing cardiometabolic diseases. The study also provides insights into the etiological associations between BMI and cardiometabolic traits. These findings provide valuable information for discovering environmental factors or specific genes influencing BMI and cardiometabolic traits and prompt research into more effective regimens for cardiometabolic diseases.

### 4.3. Strength and Limitations

This study has several strengths. This study was based on precise BMI and biochemical trait measurements, while self-report data were widely used for previous research on twin studies. Furthermore, to our knowledge, this is the first study to comprehensively analyze the genetic and environmental correlations between BMI and eight cardiometabolic factors and how they change with aging, providing evidence for genomics and environmental studies.

Our study has several limitations. First, heritability may differ between different ethnicities. Although all previous studies have reported these phenotypes as highly heritable, further confirmation is needed in other ethnic groups [52,53]. Second, in vitro and in vivo functional analyses were not conducted to validate the impact of genes and environmental factors, which is required in future research.

## 5. Conclusions

We found that BMI and cardiometabolic risk traits, including HbA1c, FBG, SBP, DBP, TC, TG, LDL-C, and HDL-C, are under moderate to substantial genetic control. The total genetic contribution for all these phenotypes presented decreasing trends with ageing. Our study further provided new insights into the correlations between BMI and cardiometabolic traits and indicated that they could partly be explained by common genes and shared environmental effects. For SBP, DBP, TC, TG, and HDL, genetic correlations with BMI decrease with age. Environmental correlations with BMI displayed a significant increasing trend for HbA1c, SBP, DBP, and HDL-C as age increased. Our study revealed the role of genetic and environmental factors underlying BMI and cardiometabolic risk traits and their correlations.

## Figures and Tables

**Figure 1 nutrients-15-00164-f001:**
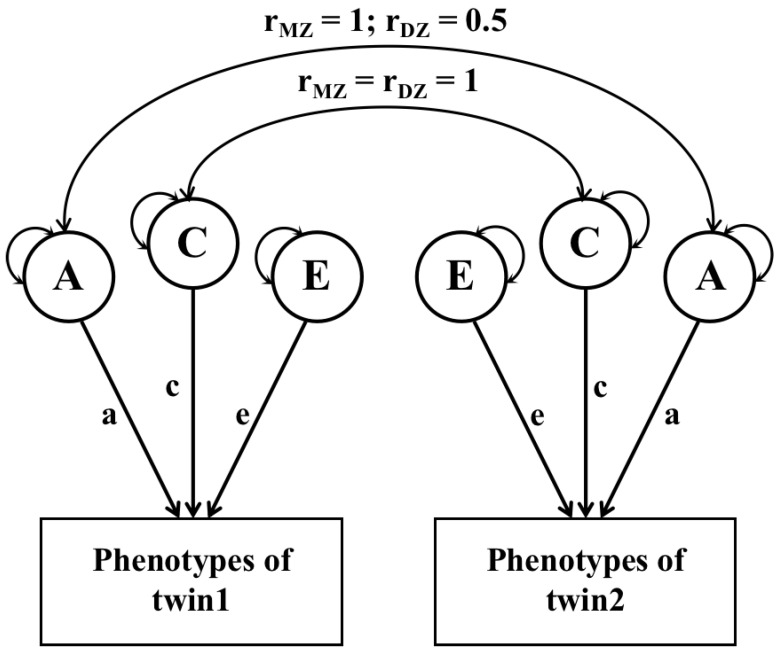
Path diagram for the univariate structural equation model. The correlations of additive genetic variance (A) are 1 for MZ and 0.5 for DZ twins. The correlations of common environmental variance (C) are 1 for both MZ and DZ twins. Unique environmental variance (E) is always uncorrelated. A, additive genetic influence; C, shared environmental influence; E, unique environmental influence; a, c, e are path coefficients of the variances.

**Figure 2 nutrients-15-00164-f002:**
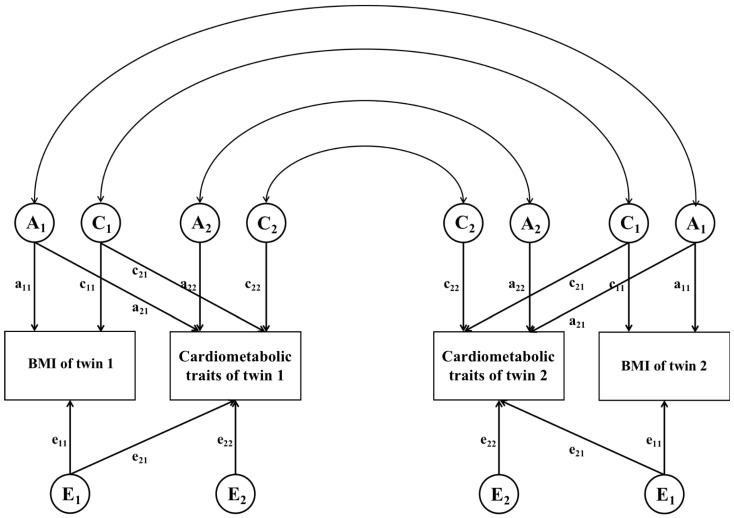
Path diagram for the bivariate structural equation model. A, additive genetic influence; C, shared environmental influence; E, unique environmental influence; a_11_, c_11_ and e_11_, unique effects on BMI; a_22_, c_22_ and e_22_, unique effects on cardiometabolic traits; a_21_, c_21_ and e_21_, common effects between BMI and cardiometabolic traits.

**Figure 3 nutrients-15-00164-f003:**
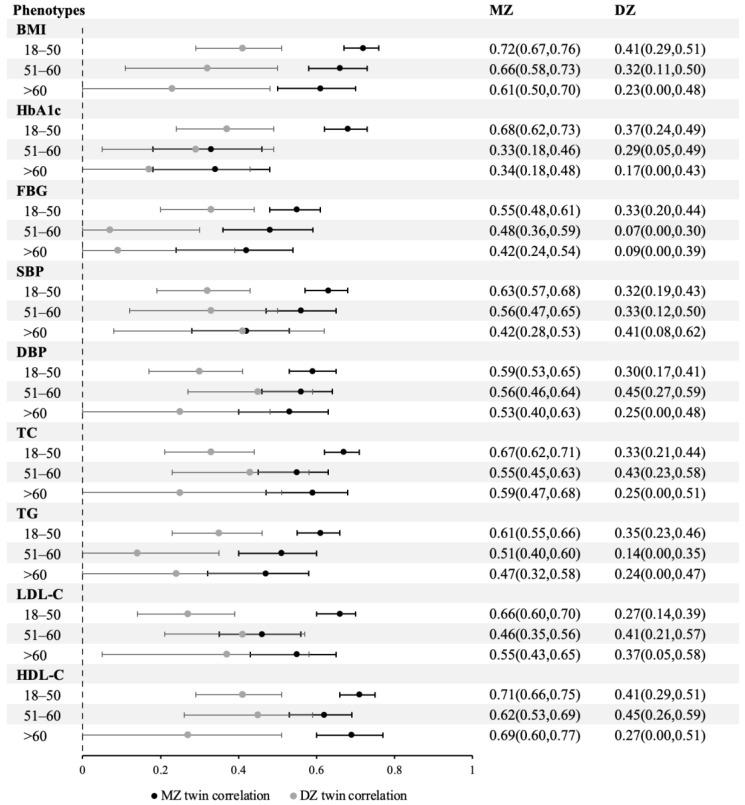
Twin correlations for BMI and cardiometabolic traits by age group. Abbreviations: MZ, monozygotic; DZ, dizygotic; BMI, body mass index; HbA1c, glycated hemoglobin; FBG, fasting blood glucose; SBP, systolic blood pressure; DBP, diastolic blood pressure; TC, total cholesterol; TG, plasma triglyceride; LDL-C, low-density lipoprotein cholesterol; HDL-C, high-density lipoprotein cholesterol.

**Figure 4 nutrients-15-00164-f004:**
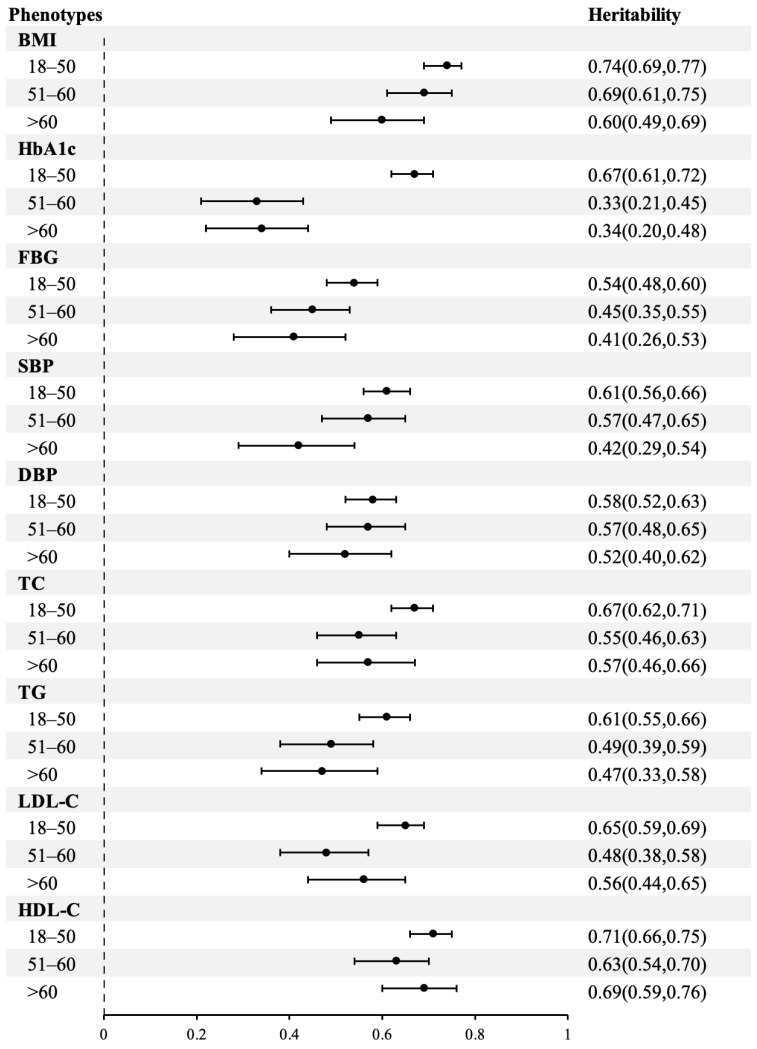
Heritability for BMI and cardiometabolic traits by age group. Abbreviations: BMI, body mass index; HbA1c, glycated hemoglobin; FBG, fasting blood glucose; SBP, systolic blood pressure; DBP, diastolic blood pressure; TC, total cholesterol; TG, plasma triglyceride; LDL-C, low-density lipoprotein cholesterol; HDL-C, high-density lipoprotein cholesterol.

**Figure 5 nutrients-15-00164-f005:**
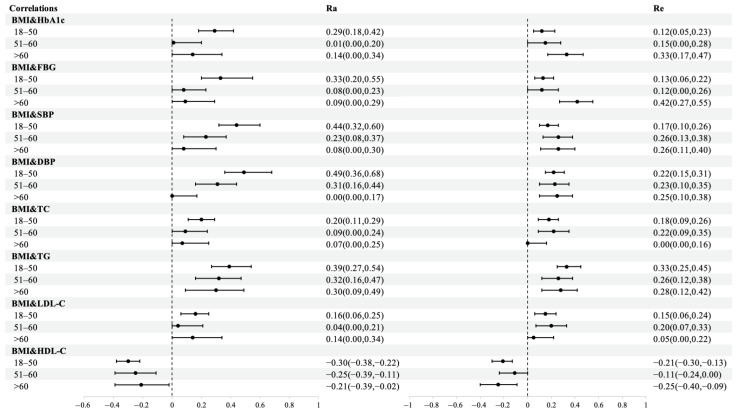
Genetic and environmental correlations between BMI and cardiometabolic traits by age group. Abbreviations: Ra, genetic correlations; Re, environmental correlations; BMI, body mass index; HbA1c, glycated hemoglobin; FBG, fasting blood glucose; SBP, systolic blood pressure; DBP, diastolic blood pressure; TC, total cholesterol; TG, plasma triglyceride; LDL-C, low-density lipoprotein cholesterol; HDL-C, high-density lipoprotein cholesterol.

**Table 1 nutrients-15-00164-t001:** Characteristics of participants by age group.

	Total	18 to 50 Years	51 to 60 Years	>60 Years
N	2842	1684	702	456
Age, years *	48 (39–55)	42 (34–47)	55 (52–57)	65 (62–69)
Female, *n* (%)	978 (34.4)	632 (37.5)	230 (32.8)	114 (25.0)
MZ, *n* (%)	1690 (59.4)	942 (55.9)	438 (62.4)	306 (67.1)
BMI (kg/m^2^) *	24.6 (22.3–26.9)	25.0 (22.5–27.5)	24.4 (22.4–26.7)	23.5 (21.5–25.6)
HbA1c (%)	5.8 ± 1.2	5.6 ± 1.1	6.1 ± 1.4	6.1 ± 1.3
FBG (mmol/L)	5.8 ± 2.1	5.6 ± 1.8	6.3 ± 2.6	6.1 ± 2.1
SBP (mmHg)	133.8 ± 22.2	127.8 ± 19.5	138.9 ± 22.3	148.2 ± 23.2
DBP (mmHg)	82.2 ± 13.8	80.3 ± 13.6	85.1 ± 13.8	85.0 ± 13.4
TC (mmol/L)	4.9 ± 1.0	4.8 ± 1.0	5.1 ± 1.1	4.8 ± 0.9
TG (mmol/L)	1.9 ± 2.4	2.0 ± 2.5	1.9 ± 2.6	1.5 ± 1.1
LDL (mmol/L)	2.6 ± 0.8	2.5 ± 0.8	2.7 ± 0.8	2.5 ± 0.8
HDL (mmol/L)	1.3 ± 0.4	1.3 ± 0.3	1.3 ± 0.4	1.4 ± 0.4
Hypertension, *n* (%)	519 (18.3)	180 (10.7)	171 (24.3)	168 (36.8)
T2DM, *n* (%)	225 (7.9)	70 (4.2)	99 (14.1)	56 (12.2)
CHD, *n* (%)	61 (2.1)	15 (0.9)	28 (4.0)	18 (3.9)
Use of antihypertensive medication, *n* (%)	433 (15.2)	107 (6.4)	152 (21.7)	174 (38.2)
Use of glucoregulatory medicine, *n* (%)	199 (7.0)	60 (3.6)	77 (11.0)	62 (13.6)
Use of lipid medicine, *n* (%)	21 (0.7)	7 (0.4)	9 (1.2)	5 (1.1)

Abbreviations: MZ, monozygotic; BMI, body mass index; HbA1c, glycated hemoglobin; FBG, fasting blood glucose; SBP, systolic blood pressure; DBP, diastolic blood pressure; TC, total cholesterol; TG, plasma triglyceride; LDL-C, low-density lipoprotein cholesterol; HDL-C, high-density lipoprotein cholesterol. * Median and interquartile range.

**Table 2 nutrients-15-00164-t002:** Parameter estimates and 95% CI in best-fitting univariate models for BMI and cardiometabolic traits in full analysis population.

	Variance Component		
Phenotypes	h^2^	c^2^	e^2^
BMI	0.72 (0.69, 0.75)	0.00 (0.00, 0.00)	0.28 (0.25, 0.31)
HbA1c	0.30 (0.09, 0.54)	0.20 (0.00, 0.40)	0.50 (0.44, 0.55)
FBG	0.49 (0.44, 0.54)	0.00 (0.00, 0.00)	0.51 (0.46, 0.56)
SBP	0.39 (0.20, 0.59)	0.17 (0.00, 0.35)	0.44 (0.40, 0.49)
DBP	0.39 (0.21, 0.59)	0.19 (0.00, 0.36)	0.42 (0.38, 0.47)
TC	0.63 (0.59, 0.67)	0.00 (0.00, 0.00)	0.37 (0.33, 0.41)
TG	0.58 (0.53, 0.62)	0.00 (0.00, 0.00)	0.42 (0.38, 0.47)
LDL-C	0.60 (0.55, 0.64)	0.00 (0.00, 0.00)	0.40 (0.36, 0.45)
HDL-C	0.69 (0.65, 0.72)	0.00 (0.00, 0.00)	0.31 (0.28, 0.35)

Abbreviations: h^2^, heritability; c^2^, common environmental variance component; e^2^, unique environmental variance component; BMI, body mass index; HbA1c, glycated hemoglobin; FBG, fasting blood glucose; SBP, systolic blood pressure; DBP, diastolic blood pressure; TC, total cholesterol; TG, plasma triglyceride; LDL-C, low-density lipoprotein cholesterol; HDL-C, high-density lipoprotein cholesterol.

**Table 3 nutrients-15-00164-t003:** Phenotypic, genetic, and environmental correlations between BMI and cardiometabolic traits from the best-fitting bivariate models in full analysis population.

Correlations	R_ph_	R_a_	R_e_	P_a_	P_e_
BMI&HbA1c	0.16 (0.11, 0.21)	0.23 (0.10, 0.44)	0.14 (0.08, 0.19)	0.60 (0.35, 0.76)	0.40 (0.24, 0.65)
BMI&FBG	0.16 (0.12, 0.21)	0.18 (0.09, 0.30)	0.17 (0.11, 0.23)	0.59 (0.37, 0.74)	0.41 (0.26, 0.63)
BMI&SBP	0.21 (0.17, 0.26)	0.31 (0.19, 0.50)	0.17 (0.12, 0.22)	0.64 (0.49, 0.75)	0.36 (0.25, 0.51)
BMI&DBP	0.27 (0.23, 0.32)	0.39 (0.27, 0.55)	0.20 (0.15, 0.26)	0.69 (0.58, 0.77)	0.31 (0.23, 0.42)
BMI&TC	0.16 (0.11, 0.20)	0.19 (0.10, 0.29)	0.13 (0.07, 0.19)	0.69 (0.50, 0.83)	0.31 (0.17, 0.50)
BMI&TG	0.33 (0.29, 0.38)	0.37 (0.28, 0.48)	0.31 (0.24, 0.39)	0.66 (0.56, 0.74)	0.34 (0.26, 0.44)
BMI&LDL-C	0.14 (0.09, 0.18)	0.14 (0.06, 0.23)	0.13 (0.07, 0.21)	0.65 (0.39, 0.82)	0.35 (0.18, 0.61)
BMI&HDL-C	−0.27 (−0.31, −0.22)	−0.34 (−0.44, −0.26)	−0.15 (−0.21, −0.09)	0.80 (0.71, 0.88)	0.20 (0.12, 0.29)

Abbreviations: R_ph_, phenotypic correlations; R_a_, genetic correlations; R_e_, environmental correlations; P_a_, proportions of genetic correlations to total phenotypic correlations, which equals to R_a_/R_ph_ × 100%; P_e_, proportions of environmental correlations to total phenotypic correlations, equals to R_e_/R_ph_ × 100%; BMI, body mass index; HbA1c, glycated hemoglobin; FBG, fasting blood glucose; SBP, systolic blood pressure; DBP, diastolic blood pressure; TC, total cholesterol; TG, plasma triglyceride; LDL-C, low-density lipoprotein cholesterol; HDL-C, high-density lipoprotein cholesterol.

## Data Availability

The datasets analyzed in the current study are not publicly available due to the informed consent involved stating that the information of participants will not be disclosed to a third party. But the corresponding author can provide analytic methods, data verification, and cooperative data analysis upon reasonable request.

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
