# Peer review of "Cardiometabolic Traits in Adult Twins: Heritability and BMI Impact with Age"

_nutrients, 2022, doi:10.3390/nu15010164_

Round 1

Reviewer 1 Report

This is an interesting and well-presented study where the authors evaluated the heritability of BMI and of other cardiometabolic traits showing declines in heritability with aging, and the importance of environmental factors (and thus also of their modification with aging).

I have detected only few minor points that need to be addressed.

1) I have detected several typos in page 2, where the authors have used un-necessary “-“. Please correct the following words and go through the whole text in case there are other such mistakes: aetiol-ogy.  envi-ronment, stud-ies, contri-butions, valua-ble, cardi-ometabolic, envi-ronmental.

2) Table 1: why the BMI data is presented as mean±SD? BMI typically has a non-parametric contribution. If this is the case also in your data, please present as median [interquartile range]. Also, I would round the age entries.

Author Response

We sincerely thank you for your appreciation and recognition of our work, and we will try our best to address the questions raised.

1) I have detected several typos in page 2, where the authors have used un-necessary “-“. Please correct the following words and go through the whole text in case there are other such mistakes: aetiol-ogy. envi-ronment, stud-ies, contri-butions, valua-ble, cardi-ometabolic, envi-ronmental.

Response: Thank you for this comment. We are very sorry that we do not understand what the reviewer means by “un-necessary ‘-’”, in the text we uploaded, these words mentioned by the reviewer were not included the “-”. We guess that these typos may arise when the PDF is automatically generated.

2) Table 1: why the BMI data is presented as mean±SD? BMI typically has a non-parametric contribution. If this is the case also in your data, please present as median [interquartile range]. Also, I would round the age entries.

Response: We appreciate the reviewer’s suggestion. As suggested, corresponding modification has been made in table 1.

Thank you very much for the comments and your effort in carefully reviewing our manuscript.

Reviewer 2 Report

The authors have conducted an interesting study to assess the relative influence of genetic and environmental factors on BMI and other cardiometabolic traits and the changes in different age groups. The 1,421 twin pairs participants were from a large-scale investigation of 61,566 twin pairs recruited from China. Univariate and bivariate structure equation models adjusting age and sex were used for the analyses. The authors noted the genetic contributions underlying correlations between BMI and SBP and DBP and TC, TG, and HDL measurement decreased as age groups increased, versus environmental correlations showing a significant increasing trend for HbA1c, SBP, DBP, and HDL. These findings contribute to the current understanding of genetic and environmental contributions to cardiometabolic risks and how their contributions have changed with age. 

The study has multiple strengths, including relatively large sample size, laboratory measurement of cardiometabolic risk factors, and rigorous statistical analyses. A few suggestions are listed below for the authors' consideration: 

  1. Could the authors provide some information regarding the participant's medical history, especially cardiovascular disease and diabetes (both type 2 and type 1), and the use of medications to manage these chronic conditions? This information would help the readers to understand the participants' characteristics better. 
  2. An additional limitation is that the authors did not account for the influences of hyperlipidemia and antidiabetic medications in the analyses. Without excluding those patients or adjusting their cardiometabolic biomarkers based on evidence, it could lead to biased estimates that did not fully reflect the influence of genetic or environmental factors. 
  3. While the bivariate SEM was adjusted for age and sex, exploring potential sex-specific differences in the analysis findings would be helpful. Sex-specific heritability for cardiometabolic risks has been reported previously, and the study has a relatively large portion of female participants. 

Author Response

We sincerely thank you for your appreciation and recognition of our work, and we will try our best to address the questions raised.

Could the authors provide some information regarding the participant's medical history, especially cardiovascular disease and diabetes (both type 2 and type 1), and the use of medications to manage these chronic conditions? This information would help the readers to understand the participants' characteristics better.

Response: We appreciate the reviewer’s suggestion. As suggested, the detailed information on medical history and the use of medications for relevant chronic diseases have been included in table 1.

An additional limitation is that the authors did not account for the influences of hyperlipidemia and antidiabetic medications in the analyses. Without excluding those patients or adjusting their cardiometabolic biomarkers based on evidence, it could lead to biased estimates that did not fully reflect the influence of genetic or environmental factors.

Response: Thank you for pointing this out. We agree with the reviewer that the use of medications may affect the outcome of the results. The use of glucoregulatory and lipid medication has been adjusted as a covariate in the models involving glycemic traits and lipid-related traits.

While the bivariate SEM was adjusted for age and sex, exploring potential sex-specific differences in the analysis findings would be helpful. Sex-specific heritability for cardiometabolic risks has been reported previously, and the study has a relatively large portion of female participants.

Response: We thank the reviewers for this insightful suggestion. We have added the subgroup analyses by sex based on both univariate and bivariate SEMs, and homogeneity tests were conducted to assess whether variance components can be equal across sex groups, which were similar to the homogeneity tests for different age groups (Supplementary Table 4, 10).

Thanks again to the reviewer for the time and effort in carefully reviewing our manuscript.